# Physical Activity Levels of Chilean Children in a National School Intervention Programme. A Quasi-Experimental Study

**DOI:** 10.3390/ijerph17124529

**Published:** 2020-06-23

**Authors:** Fernando Rodríguez-Rodríguez, Carlos Cristi-Montero, José Castro-Piñero

**Affiliations:** 1IRyS Group, School of Physical Education, Pontificia Universidad Católica de Valparaíso, Valparaíso 2340025, Chile; carlos.cristi@pucv.cl; 2GALENO Research Group, Department of Physical Education, Faculty of Education Sciences, University of Cadiz, 11003 Cadiz, Spain; jose.castro@uca.es; 3Biomedical Research and Innovation Institute of CaÅLdiz (INiBICA) Research Unit, 11009 Cadiz, Spain

**Keywords:** schools, child, physical education, recess, physical activity

## Abstract

*Background*. Recess is a great opportunity to interrupt sedentary behaviour and increase moderate-to-vigorous physical activity (MVPA) in schoolchildren. This quasi-experimental study aimed to compare the levels of physical activity (PA) during the school day of children in a school intervention programme vs. those in a control group, and to determine compliance with MVPA recommendations. *Methods.* A sample of 154 schoolchildren (6–12 years old) was obtained from several schools (70 with the intervention and 84 controls). This programme was structured with a duration of 90 min/session and performed three times/week. PA levels were recorded with triaxial accelerometers during the school day: during recess, during a PA session or physical education session (PE), and during lunchtime. No pre-intervention evaluation was performed. *Results.* The MVPA of the control group was higher than that of the intervention group during the first recess (*p* < 0.001). None of the groups complied with the recommendations for steps during the PA or PE sessions. During the PA session, sedentary time was lower and MVPA was higher, in the intervention group than in the control group. Fifty percent of the children from the intervention group complied with the MVPA recommendations, vs. 22.7% of those in the control group. *Conclusions.* The schoolchildren in the intervention group performed more MVPA than those in the control group. Future interventions could include other periods, such as recess and lunchtime, which are opportunities for improving the MVPA levels of schoolchildren.

## 1. Introduction

The benefits of physical activity (PA) for children and adolescents have been widely established [1,2]. PA during childhood has an essential impact on health in adulthood, for example, increasing bone mineral density and reducing obesity risk [3]. In response to this, the World Health Organization [4] recommended that children and adolescents from 5 to 17 years of age should perform at least 60 min of moderate-to-vigorous physical activity (MVPA) daily. Additionally, a cut-off point has been established for being active: the minimum recommendation is 13,000 steps/day for boys and 12,000 steps/day for girls [5]. Noncompliance with these recommendations categorizes children as physically inactive and increases the risk of obesity and comorbidities [6].

Physical inactivity has become frequent in children and adolescents [7]. The application and evaluation of intervention programmes to promote PA in young people should be a public health priority, mainly due to the limited effectiveness of the interventions carried out to date [8,9]. One explanation for the modest effect of existing interventions is that they have not adequately focused on the most important determinants of PA [10]. Thus, PA should include a wide range of activities in which children can participate, such as actively commuting to school, participation in sport teams, playing active games, or attending sport clubs after school [11]. In addition, it is crucial to accurately identify children’s overall PA, focusing on specific times during the school day or week [12,13].

A study that evaluated the PA level in schoolchildren—using objective methods (pedometers, accelerometers)—showed that children spend almost 70% of their time engaged in sedentary behaviour [14]. Most of their sedentary time takes place in schools [15]. Therefore, schools are an ideal environment for implementing PA interventions in children, both during structured physical education (PE) and unstructured free times [16]. The school context greatly influences children and youths and can ensure a large participation in activities for maintaining active behaviours during the school day [17]. Despite these potential advantages, schools are not able to solve the problem of physical inactivity by themselves [18].

In Chile, data from different surveys suggested that PA behaviour scored low and tended to decline as children grew older. Additionally, only 34% of children achieved the PA recommendations [19].

Based on this problematic Chilean profile background of physical inactivity and obesity in schoolchildren, the Ministry of Sports designed the EDI programme (Integral Sports Schools in English). This programme is an attempt to help mitigate the risk factors associated with a sedentary lifestyle and obesity. The aim of the EDI programme is to increase sport practices in children aged 4 to 14 years old, through educational workshops lasting between 45 and 90 min that are performed three times a week (4 to 6 years old: active kindergarten; 7 to 11 years old: sport initiation at schools; and 12 to 14 years old: sports at schools). These workshops are added to the two days of physical education, so that PA is completed every day. This intervention programme has an annual duration of eight months (April–November). The individual programmes and collective sports are carried out using a biopsychosocial strategy, to promote the balanced and harmonious development of the physical, mental, and cultural participation of girls and boys [20].

Our hypothesis before the evaluation of this programme was that an intervention dynamic and recreational would encourage a greater MVPA level during the school day. Likewise, the results obtained will make it possible to improve or reorient future national programmes, with the intention of improving the low levels of school PA. For this reason, the objectives of this study were (1) to compare the levels of PA during the school day of a group of children participating in the EDI programme vs. those of a control group and (2) to assess the impact of this intervention on compliance with the international recommendations of PA.

## 2. Materials and Methods

### 2.1. Participants

Of the 2531 children who participated in the EDI programme of the National Sports Institute of Chile, a representative sample was determined, and a sample of 524 children was obtained (confidence level 99%; margin of error 5%; population proportion of 50%). For this quasi-experimental study, stratified cluster sampling was carried out to determine the schools to be evaluated (Appendix A). The schools were assigned following simple randomization procedures through the “RandBetween” function of Microsoft^®^ Excel^®^ (Office 365^®^, Redmond, WA, USA), from the 81 schools in the programme. From a total of 81 participating schools, the sampling yielded a total of 32 schools distributed in eight regions of the country (Antofagasta, 3 schools; Coquimbo, 4 schools; Valparaíso, 5 schools; Metropolitana, 6 schools; O’Higgins, 2 schools; BioBio, 4 schools; Araucanía, 4 schools and Los Lagos, 4 schools). A total of 32 schools were assessed according to the chosen representativeness criteria. The participants’ eligibility criteria were as follows: regularly attending programme sessions or physical education classes (PE session); aged between 6 and 12 years; and not belonging to any competitive sports team. Information was collected from children of these schools via a questionnaire on PA and active behaviour, as well as via an assessment of body composition and a test of motor coordination and agility. From the original sample, 24 were excluded for not meeting the age inclusion criteria (>12 years old), and 336 declined to participate. Finally, 164 children agreed to take an accelerometer with them during the school day, according to the protocol established. The intervention group sample (participants of the programme) consisted of 80 participants (6–12 years old), of which only 70 children (51 boys and 19 girls) obtained valid accelerometer data. Ten children who did not use the device or who used it for less than 8 h/day were excluded from the study. A total of 70 children were considered in the analysis.

The control group sample consisted of 84 children (8–12 years old) with valid accelerometery data (49 boys and 35 girls). There were no missing data in this group. This group came from three schools participating in the EDI programme, but with children who did not participate in the activities programme. They belonged to the same grades, had similar school infrastructure, experienced the same weather, were tested in spring, and used the same accelerometer protocol that the intervention group used (Figure 1). Finally, the total sample of both groups was 154 children.

The age range of the groups under study was based on the fact that the EDI programme includes children of these ages. In fact, the decrease in PA levels from 7–8 years justifies the intervention at these ages [21]. Additionally, sports specialization schools begin for 12- to 13-year-olds.

### 2.2. Ethical Aspects

Informed consent for participation was obtained by the signature of the parents of the selected children. This document clearly explained the methodology to be followed, the objectives of the study, the data required and the commitment to carry the device during the established times. In addition, the confidentiality and non-invasive nature of the study were explained, so that parents understood that it did not involve any risk. This study was approved by the corresponding Ethics Committee of the Pontificia Universidad Católica de Valparaiso (CCF02052017), and we followed the standards in accordance with the Declaration of Helsinki at the 64th General Assembly, Fortaleza, Brazil (2013).

### 2.3. Instruments and Evaluations

#### 2.3.1. Anthropometry

The anthropometric characteristics were measured using a standardized protocol [22]. Body weight was measured to the nearest 0.1 kg using a digital scale (Tanita HD313, Tanita, Tokio, Japan). Height was measured with a precision of 0.1 cm using a portable stadiometer (Seca 213, Seca, Hamburg, Germany). For both measurements, the participants were instructed to remove their shoes and wear light clothing. Body mass index (BMI) was calculated as kg/m^2^. Cardio-metabolic risk was defined based on waist circumference (WC), which was taken as the minimal circumference between the iliac crest and the last rib [23,24]. A non-elastic flexible tape measure was used, with the subject standing without clothing covering the waist area. These measurements were carried out by trained and experienced researchers prior to the start of the project. All measurements were made within schools and supervised by the teachers of the programme.

#### 2.3.2. Physical Activity Measure

The level of PA was evaluated with the use of an Actigraph’s triaxial accelerometer (model wGTX3BT, Actigraph Pensacola, FL, USA). This accelerometer has been shown to have a good intraclass correlation coefficient (ICC) of 0.80, considering the specific placement, frequencies, filters and epoch length used. In addition, it is precise in the unused time definition, the valid days and population-dependent algorithms (pre-schoolers, children, adolescents, adults or older adults) [25].

The accelerometer was attached to the children’s right side between the hip and the waist with a portable elastic band, or placed directly on the belt of the pants. It was used continuously for at least 8 h/day during the school day; the data between 8:00 a.m. and 4:00 p.m. were analysed.

In the intervention group, the accelerometer was carried on days with the intervention (EDI physical activity session—PA session). At the same time, the control group’s accelerometer was worn on days with regular PE classes, ensuring that they performed the activity at the same hours and for the same time. The loss of data only allowed us to analyse the days of programme execution and physical education, and it was not possible to establish an average weekly PA. The evaluation was applied in the month of October, i.e., seven months after the beginning of the intervention.

The accelerometer data were downloaded in “.Raw” files and transformed into 10-s length “Epoch Data” for children, according to Migueles et al.’s (2017) recommendations [25]. Subsequently, the data were validated in Actilife 6.13.3 software. Afterwards, all files were exported to an Excel spreadsheet for analysis. PA was analysed according to schedules, comparing the breaks (9:30 a.m. to 9:45 a.m. and 11:15 a.m. to 11:30 a.m.), the PA session or PE session and lunch time (1:00 p.m. to 2:00 p.m.), as well as the total activity during the school day.

This accelerometer estimates sedentary time and the intensity of PA, as well as energy expenditure in Kcal and metabolic equivalents (METs). The energy expenditure was obtained through the Freedson equation [26]. In addition, the number of steps per day, steps per minute and PA levels (minutes per day) were obtained. PA was categorized into the following intensity levels: sedentary (<1.5 METs), light (1.5–3 METs), moderate (3–6 METs) and vigorous (>6 METs) [27]. The cut points were as follows: Sedentary: 0–149 CPM; Light: 150–499 CPM; Moderate: 500–3999 CPM; Vigorous: 4000–7599 CPM; Very Vigorous: 7600–∞ CPM [28]. These cut points are based on the following MET equation: METs = 2.757 ± (0.0015*CPM) − (0.08957*age) − (0.000038*CPM*age), with assumed MET thresholds of 3, 6, and 9 METs (which produce cut-point boundaries of 500, 4000, and 7600 CPM, respectively).

## 3. Intervention Programme

The specific EDI programme was structured as a comprehensive sport practice workshop, serving children between 6 and 12 years of age (25 participants per group), performed three times a week for a duration of 90 min per session. Each session consists of activities to strengthen skills for life and healthy lifestyles, and children experience a wide range of pre-sport and sport activities. These activities tend to be the children’s first experiences of one or more sport disciplines. These sessions were run by sport instructors or physical education teachers specially hired for this programme.

The activities of the programme were mainly recreational and were developed based on sports, with the aim of improving basic motor skills and fitness.

The work model focused on playful activities and games, with few rules, and the use of implements such as balls of all kinds, rings, ribbons, ropes, and hurdles, among other things. The general characteristics of the programme are shown in Table 1.

## 4. Physical Education Class

The control group only had physical education once a week (90 min per session) and was not part of the intervention. In this regard, the Chilean educational policy requires the completion of only 2 h of physical education per week. This group session consists of activities about the thematic unit called “Physical activity and health”, the main objective of which is to improve the physical condition of children. This objective is accomplished by the regular practice of physical activity focused on developing motor skills and attitudes towards fair play, leadership and self-care. These sessions are developed based on structured play and the repetition of movements for learning. Here, the teachers carry out directed activities, according to the syllabus of the subject, and there is little space for free play. The basic structure consists of a playtime for warming up the body and then teacher-directed activities, where the instruction-execution sequence is repeated. Finally, there is normally a structured return to calm.

The primary endpoint with respect to the efficacy of the programme was the proportion of children achieving the MVPA recommendations through accelerometery, in comparison to the control group. Additional analyses were performed on the MVPA during recess and school free time.

## 5. Statistical Analysis

The results of the variables are presented as the mean and standard deviation for each group. The normality of the variables was studied with the Kolmogorov–Smirnov test. For normally distributed variables from independent samples, Student’s *t* test was performed to assess the differences between the means for the intervention group and the control group. The same statistical test was also used to see the differences between the intervention and control groups, with regard to the levels of sedentariness, light PA, moderate PA, vigorous PA, MVPA and steps/min during each of the time segments described previously. No positive interaction was observed by sex. The analysis was performed for the entire school day and for each of its time periods. It was not possible to establish a weekly behaviour of physical activity, due to the loss of data from the accelerometers during the process. A value of *p* < 0.05 was defined as statistically significant. The statistical software IBM ^®^ SPSS ^®^ v21 (IBM Corp. Armonk, NY, USA) was used for the analysis.

## 6. Results

The children’s ages ranged between 7 and 12 years old (9.6 ± 1.8 years). There was no difference in the BMI of the intervention and control groups, which were 20.1 ± 2.5 kg/m^2^ and 20.6 ± 5.0 kg/m^2^, respectively (*p* = 0.281). WC was 64.6 ± 11.3 cm and 68.5 ± 10.8 cm, for the intervention and control groups, respectively (*p* = 0.011). Of the 80 children in the intervention group, 10 were excluded from the analysis for presenting missing data. All of the control groups were included in the analysis, and no missing data were recorded. Overall, no significant differences were found in the different variables, except in the percentage of participants who fulfilled the daily MVPA recommendations (Table 2). The intervention group achieved a significantly higher MVPA percentage than the participants in the control group (50% vs. 22.7%, respectively; *p* < 0.001).

Table 3 shows the levels of PA during the most active moments of the school day, which are recess, the PA session/PE session and lunchtime.

The students in the control group performed twice as much MVPA (*p* < 0.001) and took twice as many steps (*p* < 0.001) as students in the intervention group during the first recess. Additionally, sedentary time was greater in the intervention group than in the control group (*p* = 0.010).

During the second recess, schoolchildren in the control group performed more VPA (*p* = 0.001) and took more steps (*p* = 0.034) than the intervention group.

The intervention group PA was significantly higher than that during the PE session of the control group, but not at the LPA level (all *p* < 0.001). Both the time performing MVPA during the PA session and the total number of steps were higher in the intervention group than in the control group (33% (Δ 7.1 min) and 37% (Δ 955 steps), respectively) during the same period. Sedentary time was significantly lower (*p* = 0.034) in the intervention group during the PA session vs. the PE session of the control group. Finally, at lunchtime, a higher level of VPA (*p* = 0.041) and MVPA (*p* = 0.033) was identified in the control group compared to the intervention group.

With respect to the step number, analysing each block of the school day, significant differences during both recesses were observed, and the steps count was higher in the control group (*p* < 0.001 and *p* = 0.034, for the first and second recesses, respectively). However, the intervention group performed significantly more steps during the PA session (2593 for the intervention group vs. 1638 for the control group during the PE session, *p* < 0.001).

Recess, the PA session/PE session and lunchtime were the most active moments of the school day (Figure 2). Here, we can observe that the percentage of MVPA during the first recess and lunchtime was higher in the control group. However, this did not affect the difference in the total MVPA during the school day, since it was balanced with the PA session (PA/PE session), which was longer for the intervention group. The most important time for achieving a greater amount of MVPA was the PA session.

## 7. Discussion

The objective of this study was to compare the PA levels of a group of children participating in the EDI school intervention programme to those of a control group, and to assess how this intervention affected MVPA during the school day.

It was identified that the intervention group did more PA during the PA session, performed more MVPA and took more steps than the control group, obtaining a higher percentage of compliance with the daily MVPA recommendations. However, PA in the intervention group was lower during recess—a time that could be used to further improve the levels and recommendations for daily PA.

### 7.1. Physical Activity in Physical Education

Regarding PA in PE classes, the recommendation is to provide at least 50% of their PE time in MVPA [29]. Previous studies indicated that the time that students performed MVPA in PE sessions was lower than this recommendation. For example, in a study carried out in children aged 7–9 years old, only 28.6% of PE classes was spent performing MVPA [30], while another Swiss study (children aged 11.1 ± 0.6 years old) indicated that 33% of a PE class was spent performing MVPA [31]. However, higher compliance has been documented in older children aged 10–14 years, reaching 48.6% [32].

Another recent study showed that in PE class, schoolchildren spent 22% of their time performing MVPA. In addition, on days with PE class, 40% of the schoolchildren met the recommended levels, and only 24% met the recommended levels on days without PE class [33].

In our study, the MVPA time shown during the intervention PA session was 21.5 min, equivalent to 35.8% of the total class time, a low value considering the international recommendation (50% MVPA). In the case of the control group, it is even lower, since the 15.6 min per class is equivalent to 26% of the MVPA time during the session (*p* < 0.001).

These results make us think that the strategies used in PE classes are not enough to be an opportunity to improve PA levels. PE classes should be more dynamic and less structured for students, as they were in the current intervention. In this regard, the use of gamification strategies (transforming activities into games that allow the achievement of the same objectives but with greater motivation) seems to be effective for improving PA levels in schoolchildren. Indeed, a U.S. study [34] showed 55% MVPA in children in gamified activity breaks, compared with 25% during the standard intervention (*p* < 0.01).

### 7.2. Physical Activity during Recess and Free Time

It has been proposed that during recess, students can spend at least 50% of the time performing MVPA [35]. However, it has been indicated that it would be more realistic that at least 40% of the time was spent performing MVPA during each break [36]. In this regard, a Spanish longitudinal study found that <10% of schoolchildren met the recommendations during recess [37]. Another observational study conducted in Japan showed an MVPA prevalence of 19.3% in recess and free time [38].

Moreover, our results showed lower values according to the established PA recommendations, with 17.4% of the time spent in MVPA for the intervention group and 23.2% spent in MVPA for the control group. An important element to consider is that each school regulates the use of recesses and are thus able to restrict PA. This opportunity could be affected if recess were used for other purposes, such as exhibitions, ceremonies, and performances. These findings demonstrate the importance of establishing intervention programmes to improve PA levels during recess. The first recess and the lunch time positively influenced MVPA in both groups.

Regarding compliance with the number of steps, it was observed that the control group took significantly more steps during recess than the intervention group, which also translated into a greater MVPA time. Recess could be used to create activities that stimulate movement in schoolchildren, through interventions such as “active recreation” (e.g., play work), adding equipment (balls, hoops, ropes, trampolines, etc.), infrastructure or music. In the same way, a recent study showed that, compared to other interventions, interventions during recess times have the best cost-effectiveness: spending only 35 cents per student increased PA by 1.8 MET-hours per day per student [39]. These results illustrate the importance of recess times as a great opportunity to improve levels of PA in schools.

Another essential aspect to note is that the students have lunch inside the school in order to continue the rest of the day’s classes. During lunchtime, there is an opportunity for students to engage in MVPA, which stands out compared to other time blocks during the day, so much so, that the control group performed the same number of minutes of MVPA during lunchtime as during the PE session (14.3 vs. 14.2 min, respectively). Additionally, previous studies reported that lunchtime helps significantly increase the level of PA in schools [40], even as well as recess time. Together, recess and lunchtime can account for up to 30% of the daily MVPA [41]. In our study, recess and lunchtime represented 15.7 min spent in MVPA (26.1% of the recommendation) in the intervention groups and 20.9 min spent in MVPA (34.8% of the recommendation) in the control group.

### 7.3. Effects of Programmes on Physical Activity

One of the most notable and positive results in this study was that the intervention group showed greater compliance with the daily recommendations for MVPA (>60 min/day); 50% of the children complied, averaging 79.1 min/day at that intensity, unlike the control group, which had only 22.7% of the children complying with the recommendation per day.

Comparatively, a U.S. intervention study showed 41% more compliance with the MVPA recommendations per day than the control group [42]. Meanwhile, another U.S. study done in low-income schools implemented a play work programme, which obtained vigorous intensity counts (12%), significantly higher (*p* = 0.03) than those of the control schools (8%) [43]. When implementing the programme, less sedentary time was obtained in those in the programme than in the children in the control group (girls: 7.4% vs. 21.2%, respectively, *p* < 0.001, and boys: 5.8% vs. 11.1%, respectively, *p* = 0.030). Moreover, there was greater participation by the intervention group children in games on the playground than by the children in the control group, especially among girls.

In addition, several studies have shown that, compared to interventions without active breaks, interventions that include 10–15 min of PA (active breaks) during the lessons for other subjects (e.g., Math and English) significantly help to improve the number of steps [44], contributing approximately 10% of the daily MVPA [45]. Active breaks contribute to enhancing the health of school-age youth, as part of a battery of strategies to introduce PA into the daily school routine. Therefore, they are identified as opportunities to improve school PA, in addition to PE classes, recess, and lunchtime. Additionally, schools should consider not only the recommendation of >60 min of MVPA per day, but also other proposals, such as the number of daily steps (boys: 13,000 or girls: 12,000), the number of steps and >50% MVPA in PE classes and >40% MVPA during recess.

## 8. Strengths and limitations

A limitation of this study is the low number of selected cases due to the loss of valid data. Another limitation was that individual selection was not performed; rather, it was by clusters. This prevented the establishment of an average weekly physical activity that would have given a better demonstration of the effectiveness of the programme. Since a baseline assessment was not established, we cannot be absolutely certain that those differences were not before the study began. However, several strengths must be acknowledged. First, a national representation from 8 of the 15 regions of the country were included, and secondly, an objective measurement was used to record PA (accelerometry). At the same time, low-income children and public schools were included.

## 9. Conclusions

In conclusion, the children from the EDI programme performed more MVPA and steps on a school day than the children in the control group. However, this positive outcome is not enough, given that only 50% of the students in the intervention group complied with the 60 min/day MVPA recommendations. In this case, the 90-min intervention did not make a difference in the total PA time on a school day. This happens because recess and lunchtimes can help add minutes of daily PA as well. Overall, the promotion of PA at schools should include other time periods, such as recess and free lunchtimes, which are an opportunity to improve the MVPA of schoolchildren. Thus, more intervention studies that include different school periods are necessary. Finally, since there were no baseline assessments, further studies are necessary, to be sure that the differences did not previously exist.

## Figures and Tables

**Figure 1 ijerph-17-04529-f001:**
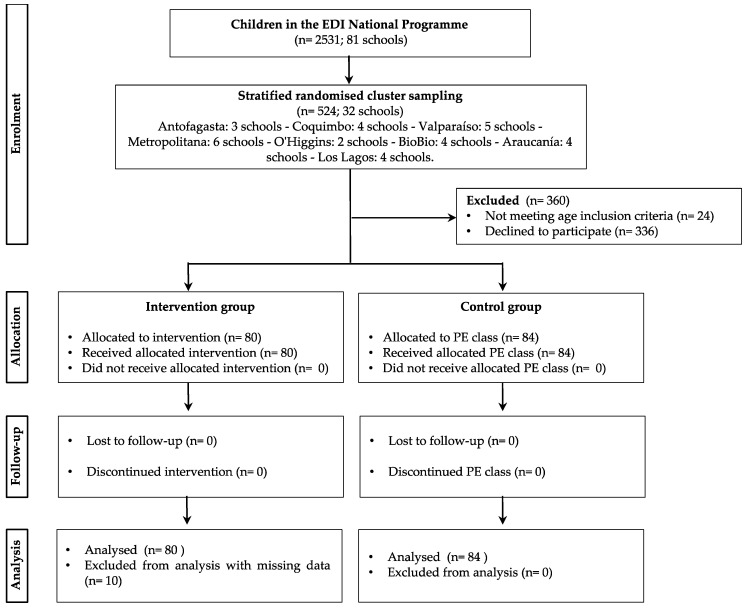
Flow diagram of the progress through the enrolment, allocation, follow-up and analysis phases.

**Figure 2 ijerph-17-04529-f002:**
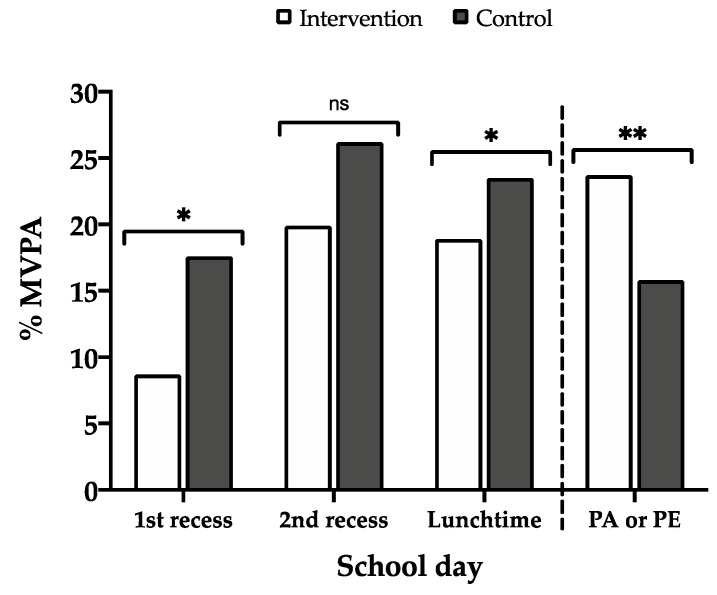
Percentage of MVPA during recess, lunchtime and the PA session/PE session for the intervention and control groups. The symbols correspond to * *p* < 0.05; ** *p* < 0.001; ns: no significant difference.

**Table 1 ijerph-17-04529-t001:** Characteristics of the programme. Component: Sport Training Schools. Subcomponent: Sport Initiation School (Annual).

Description	Each School at the Start of the Year Will Work on the Development of Basic and Specific Motor Skills (Pre- and Multi-Sport). In Addition, Activities Will Be Developed to Strengthen Life Skills (Social Abilities, Healthy Nutrition, Active Life) and Recommendations Will Be Made Regarding Healthy Lifestyles.
Beneficiaries	Approximately 25 children per school. In rural and/or isolated communities with a minimum of 15 children. Boys and girls between 6 and 12 years.
Period of Execution	8 months. April to November (Autumn, Winter and Spring).
Teacher per School	1 teacher.
Session Duration	2-session blocks of 1.5 h each.
Frequency of Sessions	2 to 3 times per week, on alternate days.
Contents	- Motor skills for locomotion, balance, and manipulation. - Development of basic physical qualities. - Basic technical foundations of a sport discipline.
Methodology	- Playful - Exploration - Analytical - Global - Mixed (Analytical–Global system)

**Table 2 ijerph-17-04529-t002:** Physical activity levels of the intervention and control groups during the school day.

	Intervention		Control		*p*-Value
	All		All		
	Mean	SD	Mean	SD	
Sedentary time (min)	257.4	±40.8	257.2	±43.9	0.981
LPA (min)	157.2	±32.4	167.4	±31.9	0.053
MPA (min)	33.7	±15.8	32.4	±11.0	0.561
VPA (min)	25.2	±14.3	22.9	±11.8	0.278
MVPA (min)	58.9	±28.1	55.3	±20.9	0.367
Sedentary time (%)	54.6	±9.5	53.6	±10.8	0.524
LPA (%)	33.6	±6.2	34.8	±6.7	0.086
MPA (%)	7.1	±3.2	6.7	±2.3	0.478
VPA (%)	5.4	±3.0	4.8	±2.5	0.432
MVPA (%)	12.5	±5.6	11.5	±4.4	0.222
Participants > 60 min MVPA (%)	50.0		22.7		<0.05
Steps/day	8467	±3118	7733	±2103	0.085

LPA = Low physical activity; MPA = Moderate physical activity; VPA = Vigorous physical activity; MVPA = Moderate-to-vigorous physical activity. Statistical significance was determined using Student’s t test with a value of *p* < 0.05.

**Table 3 ijerph-17-04529-t003:** Comparison of the levels of PA between the intervention and control groups during the school day.

PA (min)	Intervention		Control		*p*-Value
	Mean	SD	Mean	SD	
1st Recess					
Sedentary time	9.0	±2.7	7.3	±4.6	* 0.010
LPA	4.7	±2.0	5.0	±2.7	0.537
MPA	1.2	±1.3	2.3	±2.5	* 0.001
VPA	0.1	±0.2	0.3	±0.6	* 0.003
Total MVPA	1.3	±1.4	2.7	±2.8	<0.001
% MVPA in 1st recess	8.7	±9.6	17.6	±19.0	* 0.001
Step count	160	±149	369	±332	<0.001
2nd Recess					
Sedentary time	6.3	±3.4	5.0	±4.0	0.256
LPA	5.7	±2.2	6.1	±2.5	0.596
MPA	2.8	±2.7	3.5	±2.6	0.981
VPA	0.2	±0.4	0.4	±0.6	* 0.001
Total MVPA	3.0	±2.9	3.9	±2.9	0.542
% MVPA in 2nd recess	19.9	±19.4	26.2	±19.2	0.530
Step count	373	±356	469	±278	* 0.034
PA/PE session					
Sedentary time	37.0	±19.9	42.9	±12.5	* 0.034
LPA	31.6	±10.0	33.0	±8.4	0.364
MPA	18.3	±13.2	12.6	±6.3	<0.001
VPA	2.6	±2.8	1.4	±1.4	<0.001
Total MVPA	20.9	±15.4	14.2	±7.2	<0.001
% MVPA in session	23.7	±17.1	15.8	±8.0	<0.001
Step count	2593	±1684	1638	±663	<0.001
Lunchtime					
Sedentary time	23.7	±12.6	21.7	±9.4	0.289
LPA	22.7	±8.2	23.9	±6.2	0.368
MPA	10.5	±7.6	12.9	±7.3	* 0.041
VPA	0.9	±1.1	1.3	±1.8	0.076
Total MVPA	11.4	±8.2	14.3	±8.5	* 0.033
% MVPA in lunchtime	18.9	±13.7	23.5	±14.4	* 0.046
Step count	1452	±852	1630	±812	0.094

LPA = Low physical activity; MPA = Moderate physical activity; VPA = Vigorous physical activity; MVPA = Moderate-to-vigorous physical activity. * Statistical significance determined using Student’s *t* test with a value of *p* < 0.05.

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
