# Peer review of "Physical Activity Levels of Chilean Children in a National School Intervention Programme. A Quasi-Experimental Study"

_ijerph, 2020, doi:10.3390/ijerph17124529_

Round 1

Reviewer 1 Report

One relative weakness of the previous draft of the manuscript was that the language of the paper unfortunately hampered the readability. I am aware the paper has undergone some serious language editing and now reads a lot better.

I have spotted a few things that require revision but those are relatively minor. These are listed below:

Line 43: "PA should consider a wide range" - PA cannot "consider" something, only people can.

Line 47: "A study... (e.g., pedometers..." - if there was one study ("A" study) being referred to, then it doesn't make sense to use "example". I understand the authors did not mean it as such, but currently it does read that way, at least to me.

Line 69: "MORE dynamic, MORE recreational, and LESS structured..." - are there any evidence to suggest these relative wordings are justified? I think it's okay to say that the intervention was designed to accomplish these aims, but without the evidence, it might not be appropriate to say this as if it's a fact.

Line 81: This distinction between a quasi-experiment and RCT is that the former does not involve randomization. I also remember in the previous drafts this was indeed reported as an RCT. Thus the mentioning of randomization here seems a bit strange to me.

Line 84: the actual excel function is "RANDBETWEEN" (no space), I suggest to report it as such.

Line 136-139: "The accelerometer..." I am slightly confused regarding this sentence. Are the authors saying that the ICC is 0.80 if all the other parameters are held constant, or what does it mean? I suggest re-writing the sentence, or even breaking down it into shorter ones.

Line 197: "considering the sample" - what has the choice of normality test to do with the recruited sample? I do not see the connection.

Line 245: "step number" - or "step count"?

Line 270: "at least 50% of the MVPA for the day" - I believe the recommendation was that students should spend 50% of their PE time in MVPA.

Line 346: First, please see my comment above regarding randomization. Second, here it says that individual randomization was performed. But I thought the randomization was done by school - which means it was done by clusters?

Line 352: why is the inclusion of low-income and public schools a strength? Were they understudied in the past? I felt that wasn't the case.

Author Response

Comment 1.

One relative weakness of the previous draft of the manuscript was that the language of the paper unfortunately hampered the readability. I am aware the paper has undergone some serious language editing and now reads a lot better.

Response: Thank you for the comment.

I have spotted a few things that require revision but those are relatively minor. These are listed below:

Comment 2.

Line 43: "PA should consider a wide range" - PA cannot "consider" something, only people can.

Response: “consider” has been changed by “include”.

Comment 3.

Line 47: "A study... (e.g., pedometers..." - if there was one study ("A" study) being referred to, then it doesn't make sense to use "example". I understand the authors did not mean it as such, but currently it does read that way, at least to me.

Response: Thank you for the comment. “i.e.” has been deleted.

Comment 4.

Line 69: "MORE dynamic, MORE recreational, and LESS structured..." - are there any evidence to suggest these relative wordings are justified? I think it's okay to say that the intervention was designed to accomplish these aims, but without the evidence, it might not be appropriate to say this as if it's a fact.

Response: Thank you for the comment. The paragraph has been modified as: “Our hypothesis before the evaluation of this programme was that an intervention dynamic and recreational would encourage a greater MVPA level during the school day”.

Comment 5.

Line 81: This distinction between a quasi-experiment and RCT is that the former does not involve randomization. I also remember in the previous drafts this was indeed reported as an RCT. Thus the mentioning of randomization here seems a bit strange to me.

Response: Than you. The “randomised” has been deleted.

Comment 6.

Line 84: the actual excel function is "RANDBETWEEN" (no space), I suggest to report it as such.

Response: Thank you. The word has been corrected.

Comment 7.

Line 136-139: "The accelerometer..." I am slightly confused regarding this sentence. Are the authors saying that the ICC is 0.80 if all the other parameters are held constant, or what does it mean? I suggest re-writing the sentence, or even breaking down it into shorter ones.

Response: Thank you for the comment. Is true, the sentence is long. The paragraph has been corrected as: “The level of PA was evaluated with the use of an Actigraph's triaxial accelerometer (model wGTX3BT, FL, USA). This accelerometer has been shown to have a good intraclass correlation coefficient (ICC) of 0.80, considering the specific placement, frequencies, filters and epoch length used. In addition, is precise in the unused time definition, the valid days and population-dependent algorithms (pre-schoolers, children, adolescents, adults or older adults) [25]”.

Comment 8.

Line 197: "considering the sample" - what has the choice of normality test to do with the recruited sample? I do not see the connection.

Response: Thank you for the comment. The sentence “considering the sample recruited for the study”, has been deleted.

Comment 9.

Line 245: "step number" - or "step count"?

Response: Thank you. The concept has been changed to “steps count”.

Comment 10.

Line 270: "at least 50% of the MVPA for the day" - I believe the recommendation was that students should spend 50% of their PE time in MVPA.

Response: Thank you for the comment. The reference has been reviewed and corrected the sentence to “the recommendation is to provide at least 50% of their PE time in MVPA”.

Comment 11.

Line 346: First, please see my comment above regarding randomization. Second, here it says that individual randomization was performed. But I thought the randomization was done by school - which means it was done by clusters?

Response: Thank you for the comment. The sentence has been changed to “Another limitation was that not individual selection rather by clusters was performed”.

Comment 12.

Line 352: why is the inclusion of low-income and public schools a strength? Were they understudied in the past? I felt that wasn't the case.

Response: Thank you for the comment. The main reason is why few studies are done in low-income schools, especially physical activity. Unlike developed countries, public schools in Chile are more precarious and the willingness to participate in studies is very low. Studies are normally carried out in private schools where access is much easier and social conditions are more ideal.

Reviewer 2 Report

Comments to Authors:

Overall, the authors have done a very nice job with their revisions to the manuscript. The manuscript was clear and focused, and the inclusion of key figures and tables proved to be very helpful. 

Introduction Section: The introduction is clear and focused. It offers a clear overarching emphasis on the content of the paper.

Methods Section: The methods section is clear and well organized. Figure 1 (Flow Diagram) is also very helpful to provide a clear overview of the context of the study with regard to participants in the control/intervention groups.

Table 1. is also very helpful in better understanding the specifics of the programme

Results: Results section is very clear and easy to follow the researchers have done a nice job of providing a clear picture of findings between the control and intervention groups. Figure 2 is also very helpful in gaining a clearer understanding as the areas of significance as a whole between the control and intervention groups.

Discussion: The discussion section is also clear and easy to follow. The comparisons between PA and PE as well as PA outside of PE, and specifically during Recess and Free Time is appropriate. It definitely appeared that the intervention program proved to increase students overall PA levels and as is stated “greater compliance” toward daily recommendations of MVPA. The authors have done a nice job of drawing comparisons to similar research studies as well. I think it is important that the authors highlight the low number of “selected cases” as they have done within the limitations section.

Author Response

General Comments.

Overall, the authors have done a very nice job with their revisions to the manuscript. The manuscript was clear and focused, and the inclusion of key figures and tables proved to be very helpful.

Introduction Section: The introduction is clear and focused. It offers a clear overarching emphasis on the content of the paper.

Methods Section: The methods section is clear and well organized. Figure 1 (Flow Diagram) is also very helpful to provide a clear overview of the context of the study with regard to participants in the control/intervention groups.

Table 1. is also very helpful in better understanding the specifics of the programme

Results: Results section is very clear and easy to follow the researchers have done a nice job of providing a clear picture of findings between the control and intervention groups. Figure 2 is also very helpful in gaining a clearer understanding as the areas of significance as a whole between the control and intervention groups.

Discussion: The discussion section is also clear and easy to follow. The comparisons between PA and PE as well as PA outside of PE, and specifically during Recess and Free Time is appropriate. It definitely appeared that the intervention program proved to increase students overall PA levels and as is stated “greater compliance” toward daily recommendations of MVPA. The authors have done a nice job of drawing comparisons to similar research studies as well. I think it is important that the authors highlight the low number of “selected cases” as they have done within the limitations section.

Response: Thank you very much for your comments and for giving us the opportunity to publish our work. It will be very helpful for other researchers in an incipient line of research in Chile.

This manuscript is a resubmission of an earlier submission. The following is a list of the peer review reports and author responses from that submission.

Round 1

Reviewer 1 Report

See the attached word document for specific comments regarding the manuscript. 

Author Response

  1. Reviewer 1

Overall Comments:

The authors have conducted a very thorough research study on a very pertinent and imnportant issue. The study itself is thorough, and a lot of data has been collected. I would conduct a thorough proof-reading of the manuscript to ensure sentences are clear. The research study was an interesting read, and one that I think would be very beneficial to highlight the importance of keeping children actively engaged throughout the school day.

Response. Thank you very much for your comment and interesting in review this document.

Specific Comments to Authors

Line 42: Consider adding “can” to (in which children ‘can’ participate)

Response. Has been added “can” in the sentence.

Line 42: Consider changing “active” to “actively’ (such as “actively” commuting to school)

Response. Has been changed active to actively.

Line 46: Consider revising sentence to the following: “showed that children spend almost 70% of their time engaged in sedentary behaviour”

Response. Thank you, the sentence has been changed.

Line 47: Add “of” to “And most of their sedentary…”

Response. Added “of” in the sentence.

Line 54: Consider revising portion of sentence to “tended to decline as children grew older”

Response. Thank you, the sentence has been changed.

Line 59: “EDI’s am is to…”

Response. missing apostrophe has been added.

Line 64: Change “was developed” to “were developed”

Response. “was” has been replaced by were.

Line 65: Consider changing the word “development” as it is redundant as “developed” was used within the same sentence earlier.

Response. Thank you for the comment. “develop” has been replaced by “carried”.

Lines 67-74: Authors may consider combining the two paragraphs into one, as both are less than three sentences in length.

Response. Thank you, the paragraphs has been combined.

Comment: The authors have done a nice job with Figure 1. Clearly highlights the differences between each group.

Response. Thank you for your comment.

Line 100: Consider revising last sentence starting with “Also” a bit for clarity as it doesn’t appear to be a complete sentence.

Renponse. “Also” has been added to start the sentence.

Lines 116-119: With regard to the central obesity measurement, I question the accuracy of how WC was obtained. First, I question the validity and reliability of how WC was measured. Have previous research in this area used this method before? Second, were the individuals taking measurements appropriately trained in this technique? It would be beneficial to provide previous research in which similar techniques were used.

Response.

Thank your comment. This is an important discussion point.

First, it is necessary to clarify that the concept that is used about waist circumference is, in general, wrong, considering the WHO recommendations. This defines the measurement at the midpoint between the last rib and the iliac crest or on the umbilical point. Considering this, it should be called as “abdominal circumference”, not as “waist circumference”. This is supported by the International Society for Advances in Kineantropometry (ISAK) in its documents. WHO should review this issue.

Also, systematic reviews have been made that have not declared having controlled the measurement point.

Second point, the evidence indicates that, on the one hand, the minimum circumference is a better indicator of metabolic risk in children (Johnson et al., 2010). On the other hand, it has been defined that in the absolute measurements of WC at four commonly used anatomical locations affected obesity prevalence, the relationships with depot-specific adiposity and cardiometabolic risk factors were similar regardless of race or sex in children (Harrington, et al., 2013) .

Finally, waist circumference has not been used as a co-variable or mediator of the results in the current study, but only as a descriptive variable.

In summary, the corresponding citations supporting the use of this point were added and missing information about the evaluators was added.

Johnson, S. T., Kuk, J. L., Mackenzie, K. A., Huang, T. T., Rosychuk, R. J., & Ball, G. D. (2010). Metabolic risk varies according to waist circumference measurement site in overweight boys and girls. The Journal of pediatrics156(2), 247-252.

Stewart,A.,Marfell-Jones,M.,Olds,T.,&deRidder,H.(2011).Interna-

tional Standar ds for Anthropometr ic Assessment. Lower Hutt, New Zea-

land: International Society for the Advancement of Kinanthropometr

Stewart,A.,Marfell-Jones,M.,Olds,T.,&deRidder,H.(2011).Interna-

tional Standar ds for Anthropometr ic Assessment. Lower Hutt, New Zea-

land: International Society for the Advancement of Kinanthropometr

Harrington, D. M., Staiano, A. E., Broyles, S. T., Gupta, A. K., & Katzmarzyk, P. T. (2013). Waist circumference measurement site does not affect relationships with visceral adiposity and cardiometabolic risk factors in children. Pediatric obesity8(3), 199-206.

Line 135: Revise to “Accelerometer data were….”

Response. The sentence has been rewrite.

Comment: Table 1 is somewhat unclear in its layout. Perhaps the authors can modify sections to show clearer discretion between each other. For example instead of centering the content it can be aligned to the left. Bullet its are included at some points but not all. Perhaps bullet points could be included throughtout. Consider reviewing format guidelines for tables.

Response. Thank you for the comment, the Table 1 has been corrected.

Comment: The authors have provided a clear and appropriate Statistical Analysis section. Nice job.

Response. Thanks for your praise.

Line 179: Consider revising sentence for clarity. Perhaps the word “were” should be changed to “where”

Response. “Were” has been changed by “where”.

Comment: Table to is clear overall, but for the third line in the table make sure it goes across the entire page.

Response. Thank you, the table 2 has been improved.

Line 193: Delete “The” and start sentence with “Table”

Response. The sentence has been corrected.

Line 194: Revise ‘the recess’ to simply ‘recess’

Response. “the” has been deleted.

Lines 196-197: Instead of saying “steps counts” refer to it as “step counts”

Response. The word has been corrected.

Line 198: Change “performed a higher” to “performed at a higher…”

Response. “at” has been added.

Line 201: Please revise “at the except of the LPA level” It is unclear what is meant by “except”

Response. The sentence has been corrected.

Line 203: Delete “Besides” and start sentence with “Sedentary”

Response. “Besides has been deleted”.

Line 226: Please review the initial sentence. Starting out with “Respect” is unclear. Perhaps add “With respect…”

Response. “with” has been added.

Line227: What is meant by “appreciated”? Consider revising.

Response. “appreciated” has been changed by “observed”

Line 232: Please spell out “1st”

Response. “first” has been written.

Line 233: Consider revising “this does not affect” to “this did not affect” past-tense. Also in line 234, instead of saying “it is balanced” consider “it was balanced”

Response. Thank you, both sentences has been changed.

Line 235: Consider revising “The most important time to reach a greater amount of time in MVPA” to “The most important time to reach a greater amount of MVPA…”

Response. “time in” has been deleted.

Comment: Figure 2 should have a more defined color for the intervention groups as you cannot see the actual bar. Consider a grey color or white with a black outline.

Response. Thank you, a new graph version has been added.

Line 247: Instead of saying “different hours of the school day, as recesses and lunch time” consider revising to “across the school day.”

Response. The sentence has been corrected.

Line 251: when referring to physical activity use “PA”

Response.  Physical activity has been replaced by PA.

Line 253: Change from “Respect” to “Regarding”

Response.  Respect has been replaced by Regarding.

Line 255: Change “that” to “than”

Response.  that has been replaced by than.

Lines 255-256: The sentence starting with “For example” should be reviewed and revised for clarity.

Response.  The sentence has been rewritten.

Line 267: Consider revising “strategy” to “strategies”

Response.  strategy has been replaced by strategies.

Line 267: Delete “it”

Response.  Deleted it

Line 268-269: revise sentence to “PE classes should be more dynamic and less structured for students, as it has been in the case of the intervention.”

Response.  The sentence has been rewritten.

Line 275: Consider revising initial sentence. Perhaps “It has been proposed that during recess students are engaged in at least 50% MVPA.”

Response.  The sentence has been rewritten.

Lines 275-277: The authors are suggested to use a different word than “proposed” as it does not appear to fit within the paragraph in its current form.

It is suggested that the authors create paragraphs of a minimum of three sentences or more throughout the manuscript.

Response.  The paragraph has been rewritten.

Line 279: If the authors are referring to one study, please revise sentence. “Another study, conducted in Japan, showed a prevalence…..”

Response.  The sentence has been rewritten.

Line 282: Delete “a” and change 17,4% to 17.4%

Response.  Deleted a, and the comma by point has been changed.

Lines 291-293: The authors refer to using 35 center per student. What is meant by “using”? Please clarify this more in-depth.

Response. Using has been replaced by spending only

Line 295: Change to “During lunchtime, there is a considerable dedication to students engaging in MVPA, which stands out compared to other time blocks during the day.”

Response.  The sentence has been replaced.

Line 304: Change “result” to “results”

Response.  result has been replaced by results.

Line 306: Modify section “ averaging who complied, 79.1 at that intensity.” This was unclear.

Response.  The sentence has been rewritten.

Line 309: Instead of starting sentence out with “Other” revise and say “While another….”

Response.  The sentence has been rewritten.

Line 311: Delete “Besides”

Response.  Deleted Besides.

Line 317: Capitalize and change “maths” to “Math” also delete “etc…” since you use “e.g.,”

Response. Changes have been made.

Line 322: Instead of using the / when saying MVPA/day, it is suggested to use “per”

Response. has been changed throughout the text.

Line 324: Sentence starting with “Mentioned” should be revised as the use of this word to start the sentence is somewhat confusing. What is it that you are referring to.

Response.  The sentence has been rewritten.

Line 335: Consider revising “from EDI programme” to “from the EDI programme” and also changed “perform” to “performed”

Response.  The sentence has been rewritten.

Line 338: Modify “In this case 90-minute” to “In this case, the 90-minute intervention”

Response.  The sentence has been rewritten.

Line 342: Revise “interventional” to “intervention” and change “including” to “include”

Response. Changes have been made.

Comment: The authors should double-check the references list to ensure all citations have been included and that all references are according to appropriate formatting guidelines.

Response. The references has been reviewed.

Reviewer 2 Report

The paper under review described a study in which authors evaluated the effectiveness of an in-school physical activity program. Generally speaking, the writing was clear, but would greatly benefit from some careful proofreading (e.g., the first sentence of the abstract!). I also felt the authors could further highlight the strengths and novelty of the study, which is currently lacking in the current version of the paper. I would also appreciate a bit more information on the intervention itself. For example, how the intervention differentiates from a typical PE or activity session. More specific comments regarding the manuscript are presented below:

1. Line 35:
Why was a standard using steps per day used while the authors had access to MVPA data? I believe in the result section the authors did in fact use MVPA standards instead. So this probably needs to be revised.

2. Line 78: "The representativeness of the programme at the national level was calculated at 99% confidence, 5% error resulting in a sample of 644 participants"
I have some difficulty understanding this sentence, perhaps the authors could provide a the steps in arriving with the number of 644? If participants were further stratified, are they still representative?

3. Line 88: 
The choice of the schools for the control group seemed a little random. Could the authors provide a bit more information on how these schools were selected, and why 3, but not more schools were chosen?

4. Line 117: 
Is the WC an appropriate measure of central obesity for kids of this age? I am rather doubtful.

5. Line 126:
Why were accelerometers placed in a small bag before there were attached to the waist? 
Also, was the data for each participant taken from 8am to 4pm on a single day? 

6. Line 135:
Why were 10-sec epochs used? I may be wrong but from what I recall this is not the epoch length the Freedson cutoffs were based on?

7. Table 1: "1 chronological hour, considering the group characteristics 1.5h"
I am not sure I understand this statement.
Also, the frequency of classes is "PREFERABLY 3 times per week", so is that also the ACTUAL frequency? 

8. Line 173: 
I was not aware one could adjust for other variables using t-test. Were other types of analyses used instead?

9. Table 3:
How long are the 1st and 2nd recesses? 

10. Line 250:
In my experience (albeit only in my region), the activity levels of students are greatly affected by school policy. I must admit I do not know the situation in Chilean schools at all, but I would appreciate if the authors could explain whether this might be a case. 
If yes, would this affect the comparability of recess PA levels across the schools?

11. Section 6.1:
The authors have made a case that PE classes should provide more opportunities for students to be active. In principle, I agree with this statement. However, PE is also not ONLY about MVPA, but to provide students with the knowledge and skills to be physically active OUTSIDE school as well. I believe this should not be overlooked.

Author Response

  1. Reviewer 2

The paper under review described a study in which authors evaluated the effectiveness of an in-school physical activity program. Generally speaking, the writing was clear, but would greatly benefit from some careful proofreading (e.g., the first sentence of the abstract!). I also felt the authors could further highlight the strengths and novelty of the study, which is currently lacking in the current version of the paper. I would also appreciate a bit more information on the intervention itself. For example, how the intervention differentiates from a typical PE or activity session. More specific comments regarding the manuscript are presented below:

Response. Thank you very much for the comments. They sincerely help improve the quality of the document.

  1. Line 35:
    Why was a standard using steps per day used while the authors had access to MVPA data? I believe in the result section the authors did use MVPA standards instead. So this probably needs to be revised.

Response. Thanks for the observation. The standard of steps per day presented in this section have been shown to give a conceptual framework, but have not been put above the fulfillment of MVPA per day. The effect of the program is based on compliance with the MVPA and the steps only add additional information.

  1. Line 78: "The representativeness of the programme at the national level was calculated at 99% confidence, 5% error resulting in a sample of 644 participants"
    I have some difficulty understanding this sentence, perhaps the authors could provide the steps in arriving with the number of 644? If participants were further stratified, are they still representative?

Response. Thank you very much for your observation. This paragraph has been completely corrected to add the missing information and clarify the sample selection procedure. Also, a new flow chart has been created to better explain.

  1. Line 88: 
    The choice of the schools for the control group seemed a little random. Could the authors provide a bit more information on how these schools were selected, and why 3, but not more schools were chosen?

Response. More information has been added. But the main reason is that few schools decided to participate in the study. Furthermore, in these schools few parents authorized the participation of their children. This further reduced the final sample that was evaluated.

  1. Line 117: 
    Is the WC an appropriate measure of central obesity for kids of this age? I am rather doubtful.

Response. Of course, there are new studies that point this out. Trying to understand the question, we changed the concept to cardio-metabolic risk.

However, WC this has not been a relevant variable in the presentation of these results, therefore, it has been decided not to add more information in this regard.

  1. Line 126:
    Why were accelerometers placed in a small bag before there were attached to the waist? 
    Also, was the data for each participant taken from 8 am to 4 pm on a single day? 

Response. The accelerometer was placed on the waist, but the device has a small bag to prevent it from falling to the floor or breaking. However, this sentence has been deleted.

Respect to the second question, no, the accelerometer was worn all day, but the validated data was only between 8 am and 4 pm, which is the length of the school day.

Also, the accelerometer was used more days of the week by each group, but missing data did not allow analyzing the days of program execution and physical education, since a weekly average could not be made.

New information in the text has been added.

  1. Line 135:
    Why were 10-sec epochs used? I may be wrong but from what I recall this is not the epoch length the Freedson cutoffs were based on?

Response. Thank you for the question. We don't use the Freedson Epoch, but instead use the suggestions of Migueles et al., 2017, which points to children between 3 and 15 epoch lengths. Freedson reference was used for energy expenditure.

New information in the text has been added.

  1. Table 1: "1 chronological hour, considering the group characteristics 1.5h"
    I am not sure I understand this statement.
    Also, the frequency of classes is "PREFERABLY 3 times per week", so is that also the ACTUAL frequency? 

Response. Thank you for the comment.

To “2 class blocks, corresponding to 1.5 h” has been corrected. Also, to “2 to 3 times per week, in alternating days” has been corrected.

  1. Line 173: 
    I was not aware one could adjust for other variables using a t-test. Where other types of analyses used instead?

Response. Thank you for the comment. Indeed, no adjustments were made because there were no differences by sex. The wrong phrase has been removed.

  1. Table 3:
    How long are the 1st and 2nd recesses? 

Response. The duration was fifteen minutes each. This is described in point 2.3.2.

  1. Line 250:
    In my experience (albeit only in my region), the activity levels of students are greatly affected by school policy. I must admit I do not know the situation in Chilean schools at all, but I would appreciate it if the authors could explain whether this might be a case. 
    If yes, would this affect the comparability of recess PA levels across the schools?

Response. Of course, we agree with you. Different school policies may affect the use of recess as an opportunity for PA. A paragraph has been added to the matter under discussion.

Just to clarify, the research team remained during the school day to observe and collect information. In this regard, it was ensured that recesses were carried out effectively and did not affect the results.

  1. Section 6.1:
    The authors have made a case that PE classes should provide more opportunities for students to be active. In principle, I agree with this statement. However, PE is also not ONLY about MVPA, but to provide students with the knowledge and skills to be physically active OUTSIDE school as well. I believe this should not be overlooked.

Response. Yes of course, We fully agree with what you point out. We can teach lifestyles, ways to stay active, and other goals associated with motor development. Unfortunately, vulnerable schools have few opportunities for PA, and this is provided by the school and PE class. These vulnerable children come from unprotected families, single-parent families, where the father or mother cannot support their development, much less provide support for PA. Faced with this scenario, the PE  becomes only a means to add PA.

In higher-quality socio-economic and socio-educational contexts, a school and family environment is ensured that enhances the practice of PA. There are committed parents, a school that provides opportunities for in-school and after-school practice. Thus the PE class can develop other specific learning.

Under this reason and according to that context, the importance of PE class has been exposed in this way.

In developing countries we need to work with families and schools to provide children with opportunities in and out-of-school and to increase PA. With this it will be possible to take time away from the PE class to fulfill the PA recommendations and give it greater educational value.

Round 2

Reviewer 2 Report

I would like to thank the authors for addressing my comments made previously. I have a few additional comments, please see below:

Line 87: How were the children divided into the groups? Were they randomly assigned? Was the randomization done at a cluster level or individual level?

Figure 1: typo for "Exclude" with missing data

Line 133: It is a pity that weekly data was not obtained, because that would better demonstrate the effectiveness of the programme. I felt this might be something worth mentioning in the discussion section.

Author Response

Answer to the reviewer (Round 2)

Comments

I would like to thank the authors for addressing my comments made previously. I have a few additional comments, please see below:

Line 87: How were the children divided into the groups? Were they randomly assigned? Was the randomization done at a cluster level or individual level?

Response. Thank you for the comment. Yes, the randomization was a cluster level. A new sentence has been added about it between lines 88 to 95 to clarify.

Figure 1: typo for "Exclude" with missing data

Response. Sorry for our mistake. It has already been corrected.

Line 133: It is a pity that weekly data was not obtained because that would better demonstrate the effectiveness of the programme. I felt this might be something worth mentioning in the discussion section.

Response. Of course, it was a limitation of the study. We have been added a sentence in the Statistical Analysis section and a sentence in Strengths and limitations about it.

Round 3

Reviewer 2 Report

I was slightly rushed to complete my previous review so I only focused on the parts where the authors made changes. However, taking a closer look again, I have a few additional comments / feedback which are listed below. My apologies for missing out some of these previously.

Line 91-92: "Ten children... were excluded."
I believe they were simply excluded from the analyses and not from the program itself? 

Line 118: "Both measurements participants were instructed to remove their shoes and wearing light clothing." <- "FOR" both measurements?

Line 122: "These measurements were carried out by trained, experienced, and trained researchers..." <- "trained" was included twice

Line 133: "the time range was analyzed between 8:00 a.m. and 4:00 p.m." <- the data within the time range between 8am and 4pm were analyzed, not the time range itself.

Table 2 notes: "Statistical significance through a Student’s t-test with a value of p < 0.05 adjusted for sex." <- I believe in previous revisions it was mentioned that "adjusted for sex" isn't applicable.

Table 2 & 3: "MVPA: Moderate-Vigorous physical activity" <- MVPA was defined as "moderate to vigorous physical activity" earlier in the manuscript, please be consistent.

Line 215: "During the second recess, schoolchildren in the control group performed at a higher level of VPA" <- was there peak intensity higher or did they do more VPA? I believe it should be the latter and thus the text should be revised.

Generally speaking, I feel some English editing would be beneficial for the manuscript, as there is still room for improvement. I also suggest avoiding single-sentence paragraphs (I noticed at least two). 

Author Response

Answers to reviewer

Comments

I was slightly rushed to complete my previous review so I only focused on the parts where the authors made changes. However, taking a closer look again, I have a few additional comments / feedback which are listed below. My apologies for missing out some of these previously.

Line 91-92: "Ten children... were excluded."

I believe they were simply excluded from the analyses and not from the program itself?

Response. Thank you for the comment. Yes, they have only been excluded from the current study. A sentence has been added on line 92.

Line 118: "Both measurements participants were instructed to remove their shoes and wearing light clothing." <- "FOR" both measurements?

Response. Yes for both measurements participants were instructed to remove their shoes and wearing light clothing. Also, has been added “for” in the sentence.

Line 122: "These measurements were carried out by trained, experienced, and trained researchers..." <- "trained" was included twice

Response. Thank you for the comment. The word “trained” repeated has been removed.

Line 133: "the time range was analyzed between 8:00 a.m. and 4:00 p.m." <- the data within the time range between 8am and 4pm were analyzed, not the time range itself.

Response. Thank you for the observation. The sentence has been rewritten.

Table 2 notes: "Statistical significance through a Student’s t-test with a value of p < 0.05 adjusted for sex." <- I believe in previous revisions it was mentioned that "adjusted for sex" isn't applicable.

Response. Thank you very much for detecting that error. In effect, we passed that phrase from the basic model that we use to write the footers of tables in our articles. The sentence has been removed.

Table 2 & 3: "MVPA: Moderate-Vigorous physical activity" <- MVPA was defined as "moderate to vigorous physical activity" earlier in the manuscript, please be consistent.

Response. Thank you. The concept has been corrected.

Line 215: "During the second recess, schoolchildren in the control group performed at a higher level of VPA" <- was there peak intensity higher or did they do more VPA? I believe it should be the latter and thus the text should be revised.

Response. Yes, did they do more VPA. The sentence has been corrected.

Generally speaking, I feel some English editing would be beneficial for the manuscript, as there is still room for improvement. I also suggest avoiding single-sentence paragraphs (I noticed at least two).

Response. Thank you for the comment. We have done a complete revision of the language in British English format. We hope that it can now be read better.

Round 4

Reviewer 2 Report

Thank you for making the effort for revising the manuscript based on my suggestions.